# NET4 Modulates the Compactness of Vacuoles in *Arabidopsis thaliana*

**DOI:** 10.3390/ijms20194752

**Published:** 2019-09-25

**Authors:** Sabrina Kaiser, Ahmed Eisa, Jürgen Kleine-Vehn, David Scheuring

**Affiliations:** 1Plant Pathology, University of Kaiserslautern, 67663 Kaiserslautern, Germany; kaisers@rhrk.uni-kl.de; 2Department of Applied Genetics and Cell Biology, University of Natural Resources and Applied Life Sciences (BOKU), 1190 Vienna, Austria; Ahmed.Eisa@lmu.de (A.E.); juergen.kleine-vehn@boku.ac.at (J.K.-V.); 3Current address: Plant Biochemistry and Physiology, Ludwig-Maximilians-University Munich, 80539 Munich, Germany

**Keywords:** *Arabidopsis*, cell biology, cell size, plant growth, vacuole, actin cytoskeleton, vacuolar occupancy, compactness

## Abstract

The dimension of the plants largest organelle—the vacuole—plays a major role in defining cellular elongation rates. The morphology of the vacuole is controlled by the actin cytoskeleton, but molecular players remain largely unknown. Recently, the Networked (NET) family of membrane-associated, actin-binding proteins has been identified. Here, we show that NET4A localizes to highly constricted regions of the vacuolar membrane and contributes to vacuolar morphology. Using genetic interference, we found that deregulation of NET4 abundance increases vacuolar occupancy, and that overexpression of NET4 abundance decreases vacuolar occupancy. Our data reveal that NET4A induces more compact vacuoles, correlating with reduced cellular and organ growth in *Arabidopsis thaliana*.

## 1. Introduction

Vacuolar size correlates and contributes to cell size determination, thereby defining cellular elongation rates [1,2,3]. In animals, the cytoskeleton plays an important role for maintaining cellular shape and especially actin has been shown to regulate cell shape and thereby cell size [4,5]. Since plants are surrounded by shape-giving cell walls, the actin cytoskeleton cannot directly impact on cell shape. Nevertheless, it was demonstrated that actin filaments play a direct role in regulating structural changes of the vacuole [6,7]. The phytohormone auxin impacts on actin filament organization [8,9,10] as well as on vacuolar morphology. Here, auxin induces tonoplast constrictions [1] which in turn leads to a reduction of cellular space occupied by the vacuole [2]. Notably, genetic or pharmacological impairment of the actin–myosin system abolished auxin-induced changes of the vacuole and restored meristematic cell-size control and root organ growth in *Arabidopsis* [2].

Due to the close proximity of the vacuolar membrane (tonoplast) and actin filaments it has been suggested that there might be a direct physical connection [11]. However, little is known about involved molecular players and the employed mechanism for this connection. In a screen for GFP-fusion proteins labelling actin filaments, the plant-specific Networked (NET) family was identified [12]. NET proteins possess an actin-binding region and are membrane-associated, thus they specifically link actin filaments to cell organelles. From the 13 identified family members, NET4A and NET4B could potentially impact on the actin–vacuole interface, since NET4A has been shown to bind actin and overlaps with the tonoplast [12].

## 2. Results

### 2.1. NET4A Shows a Bead-on-a-String Pattern at the Tonoplast

To investigate the potential role of NET4 proteins in vacuolar morphology, we initially inspected NET4A::NET4A-GFP localization in root epidermal cells of *Arabidopsis thaliana* seedlings. While NET4A-GFP under its endogenous promoter (hereafter referred to as NET4A-GFP) is weakly detectable in meristematic cells [1], we noted that it is preferentially expressed in the late meristematic zone, correlating with the onset of cellular elongation (Figure 1a). The root epidermis is regularly spaced between longer atrichoblast and shorter trichoblast cells, which later differentiate into non-hair and root-hair cells [13]. *NET4A* expression seems to precede in trichoblast cells and is more tightly associated with the onset of elongation in the atrichoblast cells (Figure 1a and Appendix A). To confirm vacuolar localization of the NET4A-GFP signal, tonoplast staining using FM4-64 [14] was employed. NET4A-GFP did not only show a typical tonoplast signal and colocalization with the endocytic dye FM4-64, but also displayed a more filamentous signal distribution in the cell cortex, suggesting a dual localization (Figure 1b,c). This cortical NET4A-GFP signal (Figure 1c) strongly resembles actin filaments, which is in agreement with its in vitro actin binding capacity [12].

### 2.2. NET4A Localizes to Highly Constricted Vacuolar Membranes

NET4A-GFP distribution at the tonoplast was not uniform but showed a stronger vacuolar label in the center of the cell, correlating with regions of higher vacuolar constrictions (Figure 1d–f). Vacuoles are highly constricted in meristematic cells and dramatically increase in volume during cellular elongation [3]. In agreement, in elongating cells the NET4A-GFP accumulation at the tonoplast correlated well with the remaining membrane constrictions (Figure 1g). In line with this, elongated root cells, possessing fully expanded vacuoles with no or only little constrictions, displayed only a very faint NET4A-GFP signal (Appendix A).

To assess if NET4A-GFP accumulation indeed correlates with vacuolar constrictions, we next induced alterations in vacuolar morphology. The phytohormone auxin reduces vacuolar size by inducing smaller luminal vacuoles [1,2]. We used the synthetic auxin naphthalene acetic acid (NAA) and the auxin biosynthesis inhibitor kynurenine (kyn) to increase and decrease vacuolar constrictions in the root cells, respectively. In late meristematic cells, auxin treatment (200 nM NAA) induced vacuolar constrictions and a more uniform colocalization of NET4A-GFP with FM4-64 at the tonoplast (Figure 2a,b). On the contrary, kyn-induced reduction of vacuolar constrictions correlated with faint NET4A-GFP localization at the tonoplast (Figure 2c), being reminiscent of fully elongated cells. Auxin does not markedly interfere with NET4A-GFP intensity of already highly constricted vacuoles in meristematic cells [1], but in late meristematic cells the overall signal intensity of NET4A was detectably increased and decreased in auxin treated and deprived cells, respectively (Figure 2d).

To confirm this finding, we determined NET4A levels in response to increasing auxin concentrations by immunoblotting. Using a GFP antibody, we analyzed NET4A-GFP signal intensity in whole root extracts (Figure 2e). In a dosage-dependent manner, NAA application increased the NET4A-GFP protein amounts (Figure 2f). On the other hand, auxin application did neither elevate endogenous *NET4A* nor *NET4B* expression (Figure 2g). This indicates that auxin treatment leads to a higher NET4A protein abundance which is independent of transcriptional regulation. This set of data suggests that NET4A is recruited to highly constricted regions of the tonoplast, which possibly indirectly modulates protein levels of NET4A.

### 2.3. NET4A Impacts on Vacuolar Morphology

To investigate NET4A function in maintaining vacuolar morphology, we isolated *net4a* and *net4b* knock-out lines (Appendix A). We next investigated the NET4-dependent effect on vacuolar morphology in late meristematic cells, because they mark the onset of NET4A expression. We used the vacuolar morphology index (VMI) [1,2,3], which depicts the size of the biggest luminal vacuolar structure and is highly sensitive to reveal alterations in vacuolar morphology. Both *net4a-1* (SALK_017623) and *net4b* (SALK_056957) mutants displayed more spherical vacuoles accompanied with increased VMI (Appendix A). To confirm that this phenotype is attributable to *NET4* gene function, we analyzed two additional alleles, *net4a-3* (SALK_010530C) and *net4a-4* (SALK_083604C). Both lines showed a significant increase of the VMI (Appendix A). Thus, we concluded that the observed phenotype is linked to *NET4A* gene function. Together, this suggests that NET4 association at the constricted vacuolar membranes impact on vacuolar morphology.

Subsequently, we used *net4a-1* for crossing with *net4b*, creating a double mutant (hereafter referred to as *net4a net4b*). Although the *net4a net4b* double mutant also showed more spherical vacuoles, the VMI was largely not distinguishable from the single mutants (Figure 3b,d and Appendix A). This proposes a higher redundancy in the actin–vacuole pathway.

Next we generated 35S::NET4A-GFP (hereafter referred to as NET4A-GFP^OE^) overexpression lines to assess ectopic NET4A expression. NET4A-GFP^OE^ lines showed expression throughout the root, including the meristematic region. The subcellular characteristics of NET4A-GFP^OE^ remained, showing the filamentous signal at the cell cortex and the enhanced labelling of constricted tonoplast membranes (Appendix A). Interestingly, trichoblast cells, possessing more compact and constricted vacuoles, showed a higher signal intensity when compared to atrichoblast cells (Appendix A). To assess the effect of NET4A overexpression on vacuolar morphology, the VMI for NET4A-GFP^OE^ was determined. Markedly, NET4A overexpression lines showed more roundish vacuoles (Figure 3c), and increased VMI in comparison to the control (Figure 3d). Accordingly, we conclude that NET4A overexpression as well as *net4a* loss-of-function induces more roundish vacuoles. This finding is reminiscent to the stabilization and depolymerization of the actin cytoskeleton, which also both induce more roundish vacuoles and increase the VMI [2].

### 2.4. NET4 and Auxin Spatially Define Vacuolar Occupation within the Cell

To further assess the contribution of NET4 controlling vacuolar morphology, we used auxin to induce vacuolar constrictions. Following the exogenous application of auxin (200 nM, 20 h), vacuoles of *net4a net4b* double mutants and the NET4A-GFP^OE^ line were slightly less constricted when compared to wild type seedlings (Figure 3e–h). However, considering that untreated *net4a net4b* double mutants and the NET4A-GFP^OE^ lines showed already more roundish vacuoles and higher VMIs, the relative responses were not distinguishable from wild type seedlings (Figure 3i).

Based on our data, we conclude that NET4 activity affects vacuolar morphology, but does not have a major impact on the auxin effect on vacuolar shape.

However, when inspecting vacuoles of NET4A-GFP^OE^ we noted that vacuoles seemed more condensed around the nucleus, showing a larger distance to the plasma membrane (Figure 3j). Accordingly, we quantified vacuolar distance to the plasma membrane (PM) in the respective genotypes and revealed that NET4A-GFP^OE^ indeed affected this cellular trait (Figure 3k). Auxin treatments did not alter the PM to vacuole distance in wild type, but considerably increased PM to vacuole distance in NET4A-GFP^OE^ (Figure 3l).

Next, we assessed the relative vacuolar volume of *net4a net4b* double mutants and the NET4A-GFP^OE^ line in late meristematic root cells. For that, we recorded z-stacks from cells stained with a combination of the fluorescent dye BCECF (2′,7′-Bis-(2-carboxyethyl)-5(6)-carboxyfluorescein), accumulating in the vacuolar lumen, with propidium iodide, staining the exterior of cells [14]. Three-dimensional rendering of cellular and vacuolar volume allowed us to calculate the space the vacuole occupies in a given cell (vacuolar occupancy of the cell) [2]. In comparison to the control (Figure 3m), the occupancy of double mutant (Figure 3n) was increased and that of the overexpressor (Figure 3o) significantly decreased, respectively (Figure 3p). Furthermore, while control vacuoles showed tubular connections between clearly distinguishable substructures NET4 deregulation seems to render the vacuole less constricted and more spherical in general (Appendix A). Based on this set of data, we conclude that NET4 deregulation leads to vacuoles that are more spherical and in addition, that NET4A overexpression induces highly compact vacuoles.

### 2.5. NET4A-Dependent Compacting of the Vacuole Correlates with Reduced Cell Size and Root Organ Growth

Changes in vacuolar morphology correlate with stomata movement [15] and overall cellular elongation rates [1,3,16,17]. Therefore, we initially investigated whether NET4 proteins impact on cell size in the meristematic region (Figure 4a) [18]. This region displays not only the onset of NET4A expression, but also marks the transition of cells entering the elongation zone. While cell size in the *net4a net4b* double mutant remained unchanged in comparison to the control, NET4A-GFP^OE^ showed a significant reduction in cell length (Figure 4b). In agreement, the altered cell size determination correlated with reduced root organ growth (Figure 4c–e).

Considering that auxin treatment was additive to the NET4A effect on the compactness of vacuoles, we next assessed cell length and root length changes upon NAA treatment in the *net4a net4b* double mutant and in the NET4A-GFP^OE^ lines. Based on the untreated lines, we calculated the relative cell and root length. For this, cell length and root growth were set to 100% in untreated lines and used to normalize auxin-treated cells and roots. The atrichoblast cell length of *net4a net4b* as well as NET4A-GFP^OE^ were partially resistant to auxin (200 nM NAA) when compared to wild type (Figure 4f). NET4A overexpression also caused a reduced root growth sensitivity to exogenous auxin (125 nM NAA) (Figure 4g–i).

Taken together, we show that NET4A localizes to highly constricted regions in the vacuolar membrane and contributes to the compactness of the vacuole. We moreover show that the NET4A-induced changes in vacuolar shape also impacts on cellular and organ growth.

## 3. Discussion

The plant vacuole is essential for development and growth [19,20]. In growing cells, vacuole size and cell size are correlated [13,21,22]. Thus, a role of the vacuole in cell size determination has been proposed [1], which was recently supported by the finding that increased vacuolar volume allows for rapid cellular elongation [2,3,23]. However, the underlying forces regulating vacuolar morphology are not yet well understood.

Previously, it was shown that the structural organization and dynamics of the vacuole rely on the interaction between cytoskeleton and the tonoplast [2,24]. It has been suggested that the actin cytoskeleton plays a major role during vacuole inflation in the late meristematic region (or transition zone) shortly prior to the onset of cellular elongation. Here, it could provide the force to bring vacuolar structures into close proximity to allow for homotypic fusion events. In yeast, it has been reported that actin is enriched at the vacuole at sites where fusion occurs [25]. This involves tethering, docking and fusion processes which in turn are dependent on Rab-family GTPases, vacuolar SNAREs and homotypic fusion and vacuole protein sorting (HOPS) complex [26]. In plants, pharmacological actin interference using profilin led to the disappearance of transvacuolar strands and ceased cytoplasmatic streaming, thereby affecting vacuolar shape as well [27]. Furthermore, actin interference has been shown to inhibit vacuole fusion during stomatal opening [28].

The role of NET4 within the regulation of vacuolar morphology was characterized due to its actin binding capacity [5] and localization at the tonoplast (Figure 1). We have shown that NET4 abundance is crucial to maintaining vacuolar morphology and that overexpression leads to restriction in and loss-of-function to an increase of vacuolar occupancy, respectively (Figure 3). As shown recently, increase in vacuolar volume allows for rapid cellular elongation with relatively little de novo production of cytosolic content [3]. Thus, decreased vacuolar occupancy induced by NET4A overexpression might explain the observed cell size and root length limitations.

However, compared to the importance of the actin–myosin system [2], loss of NET4A and NET4B has only mild impacts on vacuolar morphology and cell size. Accordingly, we conclude that higher molecular complexity provides a high level of redundancy for tethering the actin cytoskeleton to the vacuolar membranes. It has been shown that NET3C interacts with VAP27, a contact site protein, which is required to form complexes with the cytoskeleton [29]. Notably, NET3C turnover seems to be modulated by actin [29]. It is hence tempting to assume that the auxin effect on the actin cytoskeleton [2] may indirectly affect NET protein turnover in a similar manner. In metazoans, a variety of adaptor proteins (e.g., spectrin and filamin) are known to provide specific contact sites for the actin cytoskeleton and various membranes [29]. However, most of these protein families are not present in plants and it needs to be seen how these molecular functions are achieved in plants.

## 4. Materials and Methods

### 4.1. NET4 Gene Accession Codes

Sequence data from this article can be found in The Arabidopsis Information Resource (TAIR; http://www.arabidopsis.org/) or GenBank/EMBL databases under the following accession numbers: NET4A (At5g58320) and NET4B (At2g30500).

### 4.2. Plant Material, Growth Conditions and DNA Constructs

*Arabidopsis thaliana*, Columbia 0 (Col-0) ecotype was used as control. The transgenic line NET4A::NET4A-GFP (NET4A-GFP) has been described previously [12]. The line 35S::NET4A-GFP (NET4A-GFP^OE^) was generated using Gateway cloning. The NET4A coding sequence was amplified via PCR using root cDNA from 8-days-old *Arabidopsis* seedlings. The primers are listed in Appendix A. The cDNA fragment was cloned into the pDONR221 (Invitrogen, Thermo Fisher Scientific, Carlsbad, CA, USA) using BP-clonase according to the manufacturer’s instructions. Then, the coding sequence was transferred from the entry vector into the destination vector pH7WG2 [30] using the LR clonase from Invitrogen. Transformation into *Arabidopsis thaliana* Col-0, using the floral-dip method was carried out as described before [31].

The insertion lines *net4a-1* (SALK_017623), *net4a-3* (SALK_010530C), *net4a-4* (SALK_083604C) and *net4b* (SALK_056957) were obtained from the Nottingham Arabidopsis Stock Centre (NASC) (Nottingham University, Nottingham, UK). The double mutant *net4a net4b* was generated by crossing *net4a-1* and *net4b.* For identification of homozygous lines, genotyping of all insertion lines was performed using the NASC-recommended primers (Appendix A). Insertion sites for *net4a-1* and *net4b* were located by sequencing within the third exon (*net4a-1*) and the promoter (*net4b*), respectively. Gene knockout of the respective NET4 transcript was shown by qRT-PCR.

Seeds were surface sterilized with ethanol, plated on solid one-half Murashige and Skoog (MS) medium, pH 5.7–5.8 (Duchefa, Haarlem, Netherlands), containing 1% (*w/v*) sucrose (Roth, Karlsruhe, Germany), 2.56 mM MES (Biomol, Hamburg, Germany) and 1% (*w/v*) Phytoagar (Duchefa), and stratified at 4 °C for 1–2 days in the dark. Seedlings were grown in vertical orientation at 20–22 °C under long day conditions (16 h light/8 h dark).

### 4.3. Chemicals and Treatments

α-Naphthaleneacetic acid (NAA) was purchased from Duchefa; L-kynurenine (kyn) from Sigma (St. Louis, MO, USA); and BCECF-AM, FM4-64, MDY-64 and propidium iodide (PI) from Life Technologies (Carlsbad, CA, USA). All chemicals except PI were dissolved in dimethyl sulfoxide (DMSO). NAA and Kyn were applied in solid one-half MS medium, the dyes BCECF-AM, FM4-64 and MDY-64 in liquid one-half MS medium, and PI in distilled water.

### 4.4. RNA Extraction and Quantitative Real Time PCR

Total RNA of seedlings was extracted using the innuPREP Plant RNA kit (analytic-jena, Jena, Germany). RNA samples were reverse transcribed using the iScript cDNA synthesis kit (Bio-Rad, Hercules, CA, USA). All steps were performed according to the manufacturer’s recommendations. qPCR was performed with the iQ SYBR Green Supermix (Bio-Rad) in a C1000 Touch Thermal Cycler equipped with a CFX96 Touch Real-Time PCR Detection System (Bio-Rad). The reaction procedure was set to a 95 °C heating step for 3 min, followed by 40 cycles of denaturing at 95 °C for 10 s, annealing at 55 °C for 30 s and elongation at 72 °C for 30 s. Target-specific and control primers used for quantification are given in Appendix A. Expression levels were normalized to the expression levels of Ubiquitin 5 (UBQ5). Relative expression ratios were calculated according to the so-called “2^−∆∆*C*t^” method [32] and statistical analyses were performed using Excel software.

### 4.5. Phenotype Analysis

The quantification of vacuolar morphology, compactness and occupancy as well as cell length was carried out using 6–7-days-old seedlings. To evaluate changes upon kynurenine or auxin treatment, seedlings were transferred to solid one-half MS medium supplemented with 2 µM Kyn or 200 nm NAA 18–22 h prior to image acquisition. For the analysis of the vacuolar morphology index, confocal sections above the nucleus of the root epidermis were acquired [1]. Calculations of the vacuolar morphology index were carried out in four late (distal) meristematic cells of atrichoblast files, as described previously [3]. For the quantification of plasma-membrane–vacuole distance, the same cells of the late meristematic region were analyzed. Therefore, the distance from the plasma membrane (PM) corner to the first tonoplast structure in diagonal reach was measured for all four corners of a cell, then, mean was calculated and considered as single value for the PM–vacuole distance per cell.

For quantifying the vacuolar occupancy, only one to two cells of the same defined meristematic region before the elongation zone were used per atrichoblast cell file. Cell length was measured in three early (proximal) meristematic cells of atrichoblast cell files (Figure 4a) as described before [18]. For measurements of signal intensity (mean gray value), a defined detector gain was used and individual cells in the late meristematic zone of the root were quantified. All confocal images taken to assess the vacuolar morphology index, PM–vacuole distance and cell length were analyzed using Fiji software (https://imagej.net/Fiji). Z-stack confocal images to assess the vacuolar occupancy of the cell were further processed using Imaris 8.4 (Bitplane) software (https://imaris.oxinst.com/).

For root growth determination, 6–7-days-old seedlings grown vertically on one-half MS medium plates were used, and root length quantified using Fiji software. To analyze auxin-dependent changes in root growth, solid one-half MS medium supplemented with 125 nm NAA was used. Statistical evaluation for all data were performed using Graphpad Prism 5 software (https://www.graphpad.com/scientific-software/prism/).

### 4.6. 3D Surface Rendering

Surface rendering for the reconstruction of cells and vacuoles was performed with Imaris 8.4 (Bitplane) as described previously [3]. Generated 3D models were used for the quantification of the vacuolar occupancy of the cell.

### 4.7. Western Blotting

Roots of 7-days-old seedlings expressing NET4A::NET4A-GFP were homogenized in liquid nitrogen and solubilized in extraction buffer (200 mM Tris pH 6.8, 400 mM DTT, 8% SDS, 40% glycerol, 0.05% bromphenol blue). After incubation at 95 °C for 5–10 min, samples were centrifuged and the proteins contained in the supernatant separated using SDS-PAGE (10% gel). For blotting a polyvinylidene difluoride (PVDF) membrane (Immobilon-P, pore size 0.45 µm, Millipore, Burlington, MA, USA) was used and after blocking with 5% skim milk powder in TBST (150 mM NaCl, 10 mM Tris/HCl pH 8.0, 0.1% Tween 20), the membrane was probed with a 1:20,000 dilution of mouse anti-GFP antibody (JL-8, Roche, Roche Holding AG, Basel, Switzerland) or mouse anti-alpha-tubulin antibody (B-511, Sigma). As secondary antibody, a 1:20,000 dilution of horseradish-peroxidase-conjugated goat anti-mouse antibody (pAB, Dianova, Hamburg, Germany) was used. Signals were detected using the SuperSignal West Pico chemiluminescent substrate detection reagent (Thermo Scientific) and quantified using Fiji software. Signal intensities of GFP were normalized to alpha-tubulin and statistical evaluation was performed using Graphpad Prism 5 software.

### 4.8. Confocal Microscopy

For live cell imaging, except when FM4-64 staining was performed, roots were mounted in PI solution (0.01 mg/mL) to counterstain cell walls. FM4-64 and MDY-64 staining of the tonoplast was performed as described before [14]. For 3D imaging, vacuoles were stained with BCECF-AM (10 µM solution in one-half MS liquid medium) for at least 1 h in the dark. For staining of auxin treated samples, the staining solution was supplemented with NAA in the respective concentration. Confocal images were acquired using either a Leica SP5 (DM6000 CS), TCS acousto-optical beam splitter confocal laser-scanning microscope, equipped with a Leica HC PL APO CS 20 × 0.70 IMM UV objective and Leica HCX PL APO CS 63 × 1.20 water-immersion objective or a Zeiss LSM880, AxioObserver SP7 confocal laser-scanning microscope, equipped with a Zeiss C-Apochromat 40×/1.2 W AutoCorr M27 water-immersion objective. Fluorescence signals of MDY-64 (Leica-System, excitation/emission 458 nm/465–550 nm; Zeiss-System, excitation/emission 458 nm/473–527 nm), GFP and BCECF (Leica, excitation/emission 488 nm/500–550 nm; Zeiss, excitation/emission 488 nm/500–571 nm), FM4-64 and PI (Leica-System, excitation/emission 561 nm/599–680 nm (FM4-64), 644–753 nm (PI); Zeiss-System, excitation/emission 543 nm/580–718 nm (PI)) were processed using Leica software LAS AF 3.1, Zeiss software ZEN 2.3 or Fiji software. For double labeling, images were acquired using sequential scan mode to avoid channel crosstalk. Z-stacks were recorded with a step size of 540 nm, resulting in approximately 25–35 single images.

## Figures and Tables

**Figure 1 ijms-20-04752-f001:**
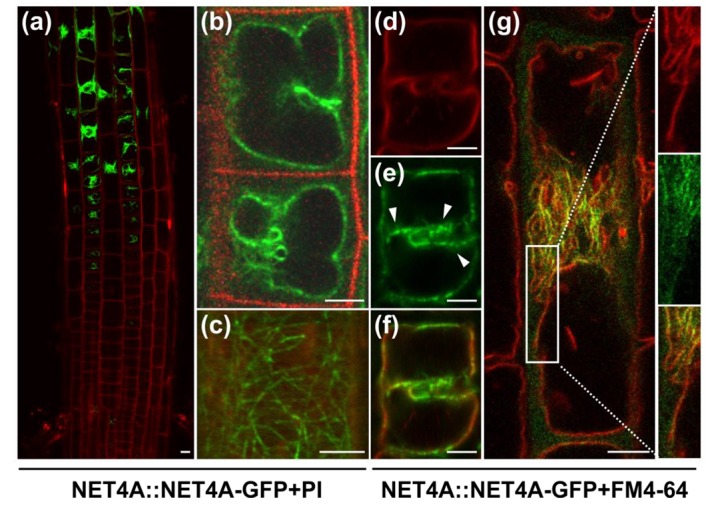
NET4A shows a bead-on-a-string pattern at the tonoplast. (**a**) NET4A-GFP expression under the endogenous promoter in *Arabidopsis* roots. Expression starts in the late meristem, initially only in trichoblast cells. (**b**) NET4A-GFP signal distribution in atrichoblast cells of the root epidermis. (**c**) Filamentous NET4A-GFP signals at the root cell cortex. (**d**–**f**) Vacuole staining by FM4-64 (3 h) shows NET4A localization preferentially at curved tonoplast areas. (**g**) NET4A-GFP signal accumulation at constrictions in elongating root cells. Detail: areas with folded tonoplast membranes show stronger NET4A-GFP accumulation in comparison to expanding areas. White arrowheads highlight punctate signals. Propidium iodide was used to stain the cell exterior, FM4-64 to stain the tonoplast. Scale bars: 5 µm.

**Figure 2 ijms-20-04752-f002:**
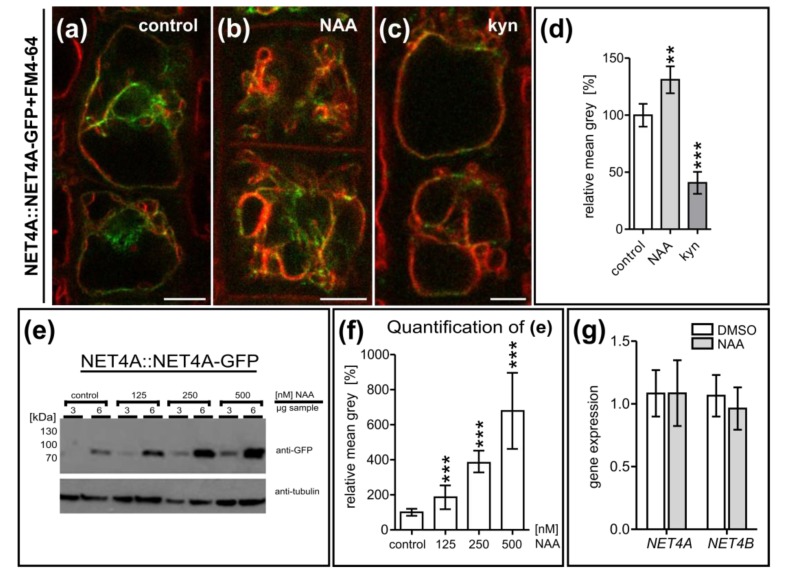
NET4A localizes to highly constricted vacuolar membranes. (**a**–**c**) Altered NET4A-GFP protein abundance upon exogenous auxin (NAA) application and depletion by the auxin biosynthesis inhibitor kynurenine (kyn). Seedlings were treated with DMSO as control (*n* = 17), 250 nM NAA (*n* = 15) and 3 µM kyn (*n* = 15). Five cells per root were considered. (**d**) Quantification of signal intensity. (**e**) Western blot analysis of NET4A-GFP (endogenous promoter) abundance upon rising NAA concentration. *Arabidopsis* root tissue was probed using a GFP-antibody. To avoid signal saturation, two different sample concentrations were loaded (3 and 6 µg). (**f**) Signal intensity of the individual NET4A-GFP bands were quantified in respect to the corresponding tubulin control bands. (**g**) To test for auxin-induced gene expression changes, qRT-PCR for both NET4 family members, *NET4A* and *NET4B*, was carried out. Error bars represent s.e.m. Student’s *t*-test, *p*-values: ** *p* < 0.01; *** *p* < 0.001. Scale bars: 5 µm.

**Figure 3 ijms-20-04752-f003:**
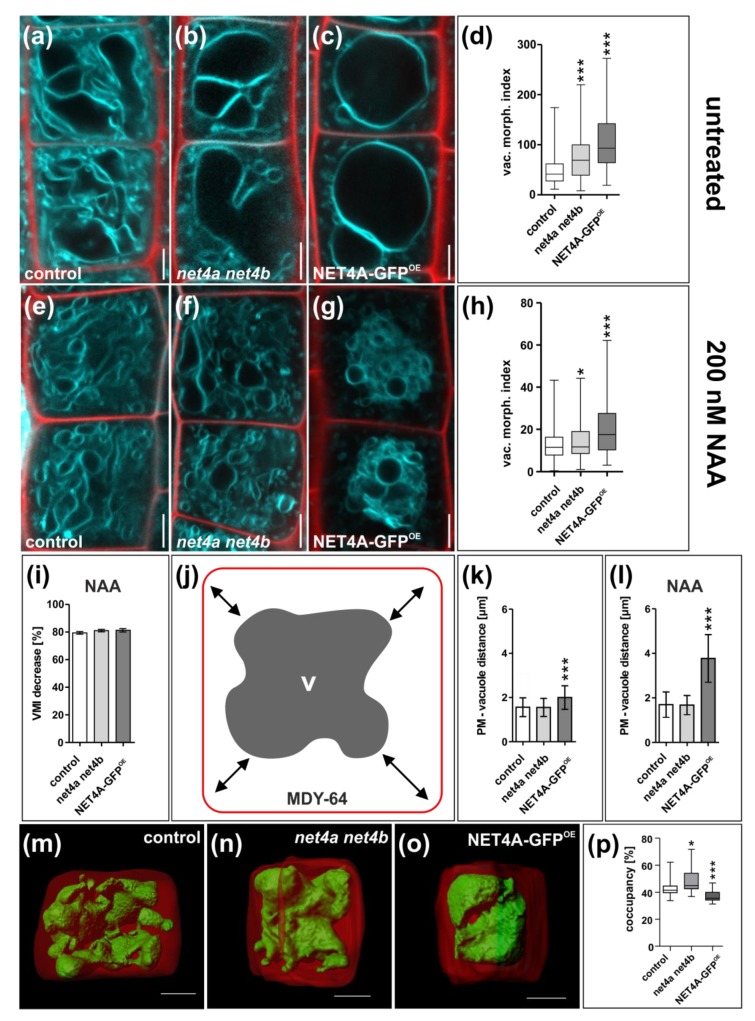
NET4 abundance impacts on vacuolar morphology, modulating compactness. (**a**–**d**) Vacuolar morphology in the late root meristem of the *net4a net4b* double knockout line (*n* = 145) and of the NET4A-GFP line driven by the 35S promoter (NET4A-GFP^OE^; *n* = 99) in comparison to the Col-0 control (*n* = 334). (**e**–**h**) Vacuolar morphology upon auxin treatment (200 nM NAA) in the control (*n* = 147), *net4a ne4b* (*n* = 153) and NET4A-GFP^OE^ (*n* = 88) line. (**i**) Relative VMI decrease. (**j**) Vacuolar compactness was assessed by measuring plasma membrane to vacuole distance based on MDY-64 staining. The distance was measured in every cell corner and the mean calculated. (**k**) Quantification of plasma membrane to vacuole distance in *net4a net4b* (*n* = 72) and NET4A-GFP^OE^ (*n* = 22) in comparison to the Col-0 control (*n* = 75). (**l**) Quantification of auxin-induced changes (200 mM, 20h) of plasma-membrane–vacuole distance in the control (*n* = 59), *net4a net4b* (*n* = 42) and NET4A-GFP^OE^ (*n* = 20). (**m**–**o**) Three-dimensional reconstructions of PI-stained cell wall (red) and BCECF-stained vacuole (green) of late meristematic cells from control (*n* = 20), *net4a net4b* (*n* = 19) and NET4A-GFP^OE^ (*n* = 22) lines. (**p**) Quantification of vacuolar occupancy of the cell. Columns of bar charts represent mean values, error bars represent s.e.m. Box limits of boxplots represent 25th percentile and 75th percentile, horizontal line represents median. Whiskers display minimum to maximum values. Student’s *t*-test, *p*-values: * *p* < 0.05; *** *p* < 0.001. Scale bars: 5 µm.

**Figure 4 ijms-20-04752-f004:**
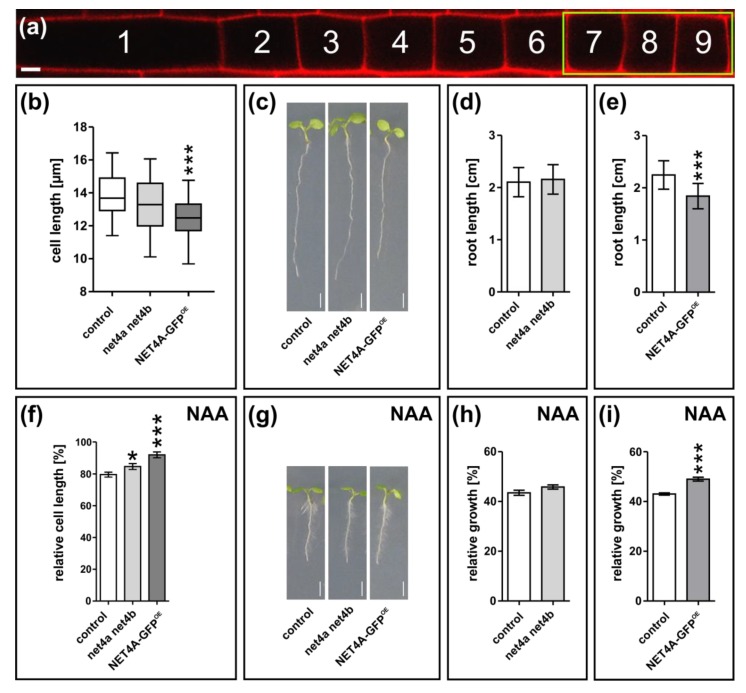
NET4A-dependent compacting of the vacuole correlates with reduced cell size and root organ growth. (**a**) Representative atrichoblast cell file for cell length quantification. The green box marks the three quantified cells per file (7 to 9). (**b**) Cell length of the *net4a net4b* (*n* = 81) double knockout line and of the NET4A-GFP line driven by the 35S promoter (NET4A-GFP^OE^; *n* = 72) in respect to the control (*n* = 147). (**c–e**) Root length of the *net4a net4b* (*n* = 27) double knockout and the NET4A-GFP^OE^ (*n* = 24) lines in respect to the corresponding control (*n* = 29 and *n* = 52). One representative experiment is shown. (**f**) Auxin-dependent (200 nM NAA, 20 h) relative cell length of the double knockout (*n* = 81) and NET4A-GFP^OE^ (*n* = 81) in comparison to the control (*n* = 147). (**g**–**i**) Relative root length of the control (*n* = 27 and *n* = 50), *net4a net4b* (*n* = 29) and NET4A-GFP^OE^ (*n* = 26) upon auxin treatment (125 nM NAA, 6 d). One representative experiment is shown. Columns of bar charts represent mean values, error bars display s.e.m. Box limits of boxplots represent 25th percentile and 75th percentile, horizontal line represents median. Whiskers display minimum to maximum values. Student’s *t*-test, *p*-values: * *p* < 0.05; *** *p* < 0.001. Scale bar: 5 µm.

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
