# Peer review of "NET4 Modulates the Compactness of Vacuoles in Arabidopsis thaliana"

_ijms, 2019, doi:10.3390/ijms20194752_

Round 1

Reviewer 1 Report

This paper refers about the role of NES4 proteins in modulation of vacuolar morphology. First the authors show co-localization of NET4A-GFP with actin filaments associated with the tonoplast, co-localizing especially with vacuole membrane constrictions. Mutants lacking functional NET4A have more spherical vacuoles, as well as NET4A-GFP overexpressing line. On the contrary, vacuolar volume of mutants was increased, but vacuolar volume of overexpressing line decreased. The authors show that the overexpression, but not the loss of NET4A, results in shorter atrichobalst cells and shorter root. Mutants and overexpression line were partially insensitive to auxin-induced root growth inhibition, suggesting that NET4A protein is part of the mechanism of auxin-induced root growth inhibition.

The experiments are very well described and documented in Figures and Supplementary Figures. NET4 has been previously shown to interact with actin filaments associated with the tonoplast. Therefore, the finding that loss of NET4A or its loss affects vacuole morphology was somehow anticipated. However, the confirmation of its function in vacuole shaping was needed and this paper is, in my opinion, clearly documenting the role of NET4A, and should be published after addressing following questions.

1. Cell size and root length experiments are loosely connected. Cell size is presented as the length of cells No. 7-8 in atrichoblast cell file (Figure 4a), which authors link with changes in root length (line 180). Did change of length of other cell types correlate with root length change? Or was the length of root tip zones (meristematic, transition, elongation) changed in overexpression mutant, which could explain shorter roots (Fig. 4e)?
2. How the relative root and cell length was calculated? What was considered as a control?
3. Relative responses of vacuolar morphology to auxin was comparable in WT and mutant line (Fig. 3I), which suggests that vacuoles were responsive to auxin. How then authors explain auxin insensitivity of mutant roots?
4. Supplementary Figure 3a: some cells seem to contain GFP signal in the nucleus. Could authors comment on it?

Author Response

Response to reviewers:

We highly appreciate the reviewers’ comments. The suggestions were very helpful and aided us to further improve our manuscript. In the revised version of the manuscript, we addressed all questions, added a new set of data (Supplemental Figure S2) and made the figure labelling more clear (Figure 1 and Figure 2). Furthermore, we explained the parts of the manuscript which raised questions in more detail and improved the discussion as suggested by the reviewers. Please, see also the detailed responses to reviewers below.

Reviewer 1

Cell size and root length experiments are loosely connected. Cell size is presented as the length of cells No. 7-8 in atrichoblast cell file (Figure 4a), which authors link with changes in root length (line 180). Did change of length of other cell types correlate with root length change? Or was the length of root tip zones (meristematic, transition, elongation) changed in overexpression mutant, which could explain shorter roots (Fig. 4e)?

We thank reviewer 1 for the positive assessment and valuable comments. We decided to measure cell length and vacuolar morphology in the meristem, because cells transit here into the elongation zone. Accordingly, these cells are an excellent system to assess changes of vacuolar morphology before and during the onset of cell expansion. In addition, NET4A-GFP under its endogenous promoter showed the strongest expression in the late root meristem (Figure 1, Supplemental Figure S1). Our previous work shows that subtle changes in this region (e.g. vacuolar morphology and cell size) correlate with defects in organ growth (Löfke et al., 2015; Scheuring et al., 2016; Dünser et al., 2019).

How the relative root and cell length was calculated? What was considered as a control?

The relative cell and root length was calculated based on the untreated respective lines. Here, growth was set to 100% and this was used to normalize auxin-treated cells and roots. That means, auxin-treated Col-0 was compared with untreated Col-0; the auxin-treated net4a net4b double mutant with the untreated double mutant and the NAA-treated NET4A-GFPOE with the untreated overexpressor line. We improved the description of our approach in the revised version of the manuscript.

Relative responses of vacuolar morphology to auxin was comparable in WT and mutant line (Fig. 3I), which suggests that vacuoles were responsive to auxin. How then authors explain auxin insensitivity of mutant roots?

Thank you again for the question. In general, upon NAA-treatment the vacuolar morphology index (VMI) remains larger for the net4a net4b double mutant and the NET4A-GFPOE (Figure 3h). It could be that, while the relative VMI response was indeed comparable in Col-0 and NET4 lines, the absolute differences (e.g. lower overall constrictions) may impact on cell expansion. On the other hand, we noticed additional auxin-induced effects in the overexpressor line which were absent in the WT. NET4A-GFPOE showed almost a doubling of the PM-vacuole distance after exogenous auxin application (Figure 3k and 3l). This shows that especially vacuoles in the NET4A-GFPOE line reacts differently to auxin than the WT. Therefore, partial auxin-resistance in roots might reflect the different behavior of vacuoles in the NET4 lines upon hormonal treatment.

Supplementary Figure 3a: some cells seem to contain GFP signal in the nucleus. Could authors comment on it?

Supplemental Figure S3a shows the Arabidopsis root epidermis, which is regularly spaced into longer atrichoblast and shorter trichoblast cells, which later differentiate into non-hair and root-hair cells. Especially in the shorter trichoblasts, the higher compactness of NET4A-GFPOE leads to the appearance of GFP signal being in the nucleus. 3D models of the overexpressor (Supplemental Movies S3) revealed spherical vacuoles which were condensed around the nucleus. Therefore, vacuoles in trichoblast cells of the NET4A-GFPOE resemble a nuclear signal distribution (although it is the tonoplast, tightly surrounding the nucleus).

Reviewer 2 Report

Plant vacuole is perhaps the largest organelle in mature plant cells, with multifunctional roles in plant growth and development. In the vegetative tissues, vacuoles act in combination with the cell wall to generate turgor, the driving force for hydraulic stiffness and cell growth. Playing crucial roles in the processes of detoxification and general cell homeostasis, vacuoles are also actively involved in cellular responses to environmental stresses. The manuscript by Kaiser et al. functionally characterized the role of NET4 genes in the regulation of the morphology and compactness of vacuoles in Arabidopsis. This is a very nice piece of work that apples molecular and cellular biology to dissect a novel component in the patterning of plant vacuoles. The presented data is clear, and the manuscript is well written. Worthy to mention is the high resolution of images and movies in the cell biology work on vacuole morphology that make the data very convincing. I support publication of this work in IJMS. A few minor concerns listed below:

An interesting finding of this study is that the NET4A-GFP signal is dramatically increased by the addition of auxin (NAA). I assume the authors used the native promoter-driven NET4A-GFP lines. Is this up-regulation due to transcriptional regulation of NET genes by auxin or does it occur at the protein level? The authors may also need to discuss more on this point considering it is an interesting point in this study. The authors showed only one allele of the net4a net4b double mutant. In the light of genetic analysis, either a second allele or transgenic complementation lines may be helpful to verify that the observed phenotypes are attributable to the genotypes/gene functions. The authors suggested NET4A-dependent compacting of the vacuole correlates with reduced cell size and root organ growth. What about other cell types? Since vacuoles are also predominantly present in mesophyll cells, does the overall growth and development of net4a net4b show any phenotypes? If not, why only in root cells? Is it because of the tissue specific expression of NET4s or other reasons? The authors may need to discuss this in more depth. Most of the findings in this study are descriptive, although they well documented the roles of NET4s in plant cells. I am wondering whether the authors could provide a proposed model or hypothesis to imagine how NET4s fulfill their function that awaits future investigation.

Author Response

Response to reviewers:

We highly appreciate the reviewers’ comments. The suggestions were very helpful and aided us to further improve our manuscript. In the revised version of the manuscript, we addressed all questions, added a new set of data (Supplemental Figure S2) and made the figure labelling more clear (Figure 1 and Figure 2). Furthermore, we explained the parts of the manuscript which raised questions in more detail and improved the discussion as suggested by the reviewers. Please, see also the detailed responses to reviewers below.

Reviewer 2

An interesting finding of this study is that the NET4A-GFP signal is dramatically increased by the addition of auxin (NAA). I assume the authors used the native promoter-driven NET4A-GFP lines. Is this up-regulation due to transcriptional regulation of NET genes by auxin or does it occur at the protein level? The authors may also need to discuss more on this point considering it is an interesting point in this study.

We thank reviewer 2 for the overall positive assessment and valuable comments. Indeed, we used the native promoter-driven NET4A-GFP line to investigate changes of NET4A abundance upon auxin treatment. We have improved the labelling of the figures (Figure 1 and Figure 2) to make more clear that we used NET4A-GFP under its endogenous promotor. Using qRT-PCR we could show that increased NET4A abundance is not due to transcriptional changes (Figure 2g), but occurs at the protein level. We followed your suggestion and discussed this section more thoroughly (line 93-96). Since NET4A is preferentially localized to curved membrane stretches, auxin-induced increase of constrictions, accompanied by recruitment of NET4A could explain the dosage-dependent increase on the protein level. However, due to the finding that net4a net4b still shows constrictions after NAA treatment, NET4 is not responsible to conduct these changes alone. Therefore, we concluded that higher molecular complexity (e.g. other proteins that can tether actin to the vacuolar membrane) provides a high level of redundancy for tethering the actin cytoskeleton to the vacuolar membranes. To discuss also this point in more detail we have extended the discussion in the revised version of this manuscript.

The authors showed only one allele of the net4a net4b double mutant. In the light of genetic analysis, either a second allele or transgenic complementation lines may be helpful to verify that the observed phenotypes are attributable to the genotypes/gene functions.

We appreciate the reviewers concerns. Originally, we have analyzed three individual (net4a-1, net4a-3 and net4a-4) NET4A alleles (see Figure below and Supplemental Figure S2). We verified the absence of NET4A gene expression by qRT-PCR-and measured the vacuolar morphology index (VMI, according to Löfke et al., 2015). In agreement with the findings for net4a-1 (Supplemental Figure S2), the other lines also showed a significant increase of the vacuolar morphology index (see Figure below). Thus, we concluded that the observed phenotype is attributable to the NET4A gene function. We used net4a-1 for crossing with net4b and focused for further analysis on the double mutant. We now included the data for net4a-3 and net4a-4 in the revised version of the manuscript.

The authors suggested NET4A-dependent compacting of the vacuole correlates with reduced cell size and root organ growth. What about other cell types? Since vacuoles are also predominantly present in mesophyll cells, does the overall growth and development of net4a net4b show any phenotypes? If not, why only in root cells? Is it because of the tissue specific expression of NET4s or other reasons? The authors may need to discuss this in more depth.

Under its native promotor, NET4A shows expression predominantly in the root and hardly in leaf mesophyll cells. Epidermis cells of the root meristem are the first cells to show NET4A expression and furthermore display the strongest signal intensity as well (Figure 1, Supplemental Figure S1). We followed your suggestion and made the choice of cell length measurements in the root meristem more clear (line 187-188). Unfortunately, the net4a net4b double mutant does not show significant growth or developmental phenotypes. Therefore, we concluded that a higher molecular complexity could provide a high level of redundancy for tethering the actin cytoskeleton to the vacuole.

Most of the findings in this study are descriptive, although they well documented the roles of NET4s in plant cells. I am wondering whether the authors could provide a proposed model or hypothesis to imagine how NET4s fulfill their function that awaits future investigation.

Thank you for pointing that out. At the moment we do not know the role of NET4 in actin-vacuole connection well enough to propose a model. However, in the revised manuscript we extended the discussion and indicated that there might be other proteins that can tether actin to the vacuolar membrane. A high level of redundancy to form cytoskeleton-organelle connections could also explain the weak net4a net4b phenotype.
